# Runge-Kutta Approximation and Decoupled Attention for Rectified Flow Inversion and Semantic Editing

## Abstract

Rectified flow (RF) models have recently demonstrated superior generative performance compared to DDIM-based diffusion models. However, in real-world applications, they suffer from two major challenges: (1) low inversion accuracy that hinders the consistency with the source image, and (2) entangled multimodal attention in diffusion transformers, which hinders precise attention control. To address the first challenge, we propose an efficient high-order inversion method for rectified flow models based on the Runge-Kutta solver of differential equations. To tackle the second challenge, we introduce Decoupled Diffusion Transformer Attention (DDTA), a novel mechanism that disentangles text and image attention inside the multimodal diffusion transformers, enabling more precise semantic control. Extensive experiments on image reconstruction and text-guided editing tasks demonstrate that our method achieves state-of-the-art performance in terms of fidelity and editability.

## 1 Introduction

Text-to-image diffusion models Ho et al. (2020); Ramesh et al. (2022); Saharia et al. (2022); Rombach et al. (2022); Balaji et al. (2023); Betker et al. (2023); Podell et al. (2024); Sauer et al. (2024); Chen et al. (2024); Esser et al. (2024) have achieved remarkable progress with powerful capabilities in generating diverse and realistic images conditioned on textual prompts. Recent research on Rectified Flow (RF) Liu et al. (2023); Lipman et al. (2023); Esser et al. (2024) models (*e.g.*, FLUX Labs (2024)) has demonstrated superior generative performance, surpassing previous DDIM-based Song et al. (2021); Dhariwal & Nichol (2021) methods such as Stable Diffusion (SD) Rombach et al. (2022); Podell et al. (2024); Sauer et al. (2024).

The image generation process in diffusion models starts from an initial Gaussian noise and progressively denoises it to approximate the target data distribution. Consequently, a critical challenge in ensuring consistency lies in *how to invert a given image to a specific noise sample that can reconstruct it faithfully*. To date, only a limited number of studies Yang et al. (2025); Rout et al. (2025); Wang et al. (2025); Deng et al. (2025) have explored inversion for RF models. However, their performance remains significantly below the VQAE Rombach et al. (2022) reconstruction upper bound. Prior work Wang et al. (2025) has demonstrated that high-order approximation can partially mitigate this degradation, motivating the development of a higher-order inversion technique with a tighter error bound tailored to RF models.

In text-guided image editing with diffusion models, the target prompt is applied during the denoising process. Therefore, another key challenge is *how to effectively leverage the source information from the inversion process to balance the trade-off between faithfulness and editability*. Existing approaches for RF models Wang et al. (2025); Deng et al. (2025); Tewel et al. (2025); Zhu et al. (2025) reuse the source attention features of query, key, and value from the inversion process to enhance the fidelity. However, state-of-the-art (SoTA) RF models adopt the Multimodal Diffusion Transformer (MM-DiT) architecture Peebles & Xie (2023), which jointly encodes and processes both text and image modalities within a unified transformer framework. As a result, directly reusing the attention features with entangled text and image information may lead to performance degradation in editing precision.

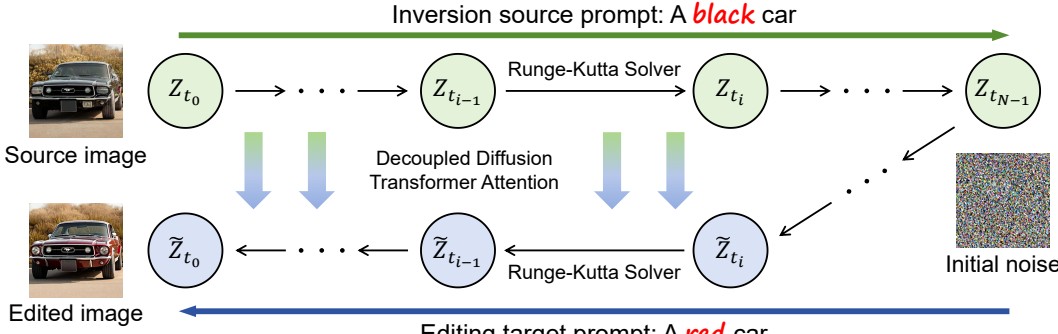

Figure 1: Conceptual illustration of our text-guided semantic editing framework.

To address the first challenge, we propose an efficient high-order inversion method for rectified flow models based on the Runge-Kutta solver of differential equations. To tackle the second challenge, inspired by DiTCtrl Cai et al. (2025), we introduce Decoupled Diffusion Transformer Attention (DDTA), a novel mechanism that disentangles text and image attention inside the multimodal diffusion transformers, enabling more precise semantic control. Extensive experiments on image reconstruction and text-guided editing tasks demonstrate that our method achieves state-of-the-art performance in terms of fidelity and editability.

As shown in Fig. 1, in this paper, we introduce Runge-Kutta (RK) approximation and decoupled attention for RF inversion and semantic editing. Specifically, to address the first challenge, we incorporate the RK method from numerical analysis into the RF sampling process and propose a high-order solver for the differential process of RF. To tackle the second challenge, we delve into the internal structure of MM-DiTs and decouple the tightly entangled text and image attention in MM-DiTs, thereby enabling more precise control over the semantic editability. To comprehensively evaluate the proposed method, we conduct extensive experiments on both image reconstruction and text-guided semantic editing tasks. Experimental results on the reconstruction task show that our RK solver improves inversion fidelity, achieving a Peak Signal-to-Noise Ratio (PSNR) gain up to 2.39 dB, which is quite significant. The results of the editing task demonstrate that our method achieves a superior overall performance in terms of fidelity and editability compared to SoTA RF-based methods.

## 2 RELATED WORK AND UNIQUE CONTRIBUTIONS

Our work is closely related to existing inversion methods and text-guided semantic editing. In this section, we first review existing work on these two related research areas. We then summarize the unique contributions of this work.

### 2.1 INVERSION FOR RECTIFIED FLOW MODELS

Inversion serves as a fundamental building block for real-world image manipulation, which has been widely studied in SD literature Mokady et al. (2023); Dong et al. (2023); Duan et al. (2024); Pan et al. (2023); Wang et al. (2025); Rout et al. (2025); Deng et al. (2025); Dhariwal & Nichol (2021); Song et al. (2021); Wallace et al. (2023); Zhang et al. (2024); Wang et al. (2024); Samuel et al. (2025). However, due to the theoretical differences between DDIM and RF, these successful DDIM-based methods cannot be directly applied to RF-based models. RF-Prior Yang et al. (2025) performs score distillation to invert a given image using RF models. However, this method incurs substantial computational overhead due to the large number of optimization steps required. RF inversion Rout et al. (2025) improves the inversion quality by employing dynamic optimal control derived from linear quadratic regulators. RF-Solver Wang et al. (2025) uses the Taylor expansion to reduce inversion errors in the ordinary differential equations (ODEs) of RF models. FireFlow Deng et al. (2025) reuses intermediate velocity approximations to achieve the second-order accuracy while maintaining the computational cost of a first-order method. Unlike previous methods, this work leverages the RK method to provide a higher-order, training-free, and more accurate approximation of the differential

process in RF models. Furthermore, we conduct comprehensive experiments to identify the optimal Butcher tableau configuration for each solver order, ensuring the best reconstruction accuracy using RF models.

## 2.2 Text-guided Semantic Editing with DiT-based models

Image editing aims to modify the visual content in a controllable manner while preserving the overall structure of the original image. Among various approaches, text-guided semantic editing has attracted the most research attention due to its remarkable flexibility Huang et al. (2025). Text-guided semantic editing based on diffusion models has been widely studied in recent years Kim et al. (2022); Miyake et al. (2023); Hertz et al. (2023); Tumanyan et al. (2023); Mokady et al. (2023); Dong et al. (2023); Brooks et al. (2023); Parmar et al. (2023); Kawar et al. (2023); Wang et al. (2023); Pan et al. (2023); Wallace et al. (2023); Huang et al. (2024). However, due to the significant structural differences between UNet-based (*e.g.*, SD) and DiT-based (*e.g.*, FLUX) models, these methods fail to apply to DiT-based models directly. RF-Solver Wang et al. (2025) and Fire-Flow Deng et al. (2025) replace the value attention feature of single-stream DiT blocks in the editing branch with those from the inversion branch to balance the faithfulness and editability. KV-Edit Zhu et al. (2025) caches the keys and values corresponding to the background during the inversion process and reuses them during the denoising to improve background consistency. However, it requires an additional mask input to separate the foreground from the background, making it less flexible than purely text-guided semantic editing. Add-it Tewel et al. (2025) utilizes both keys and values of DiT blocks from the source image to guide the editing process. However, its latent blending mechanism relies on SAM-2 Ravi et al. (2025) to obtain an object mask, introducing additional computational overhead. In contrast to previous methods, this work delves into the internal structure of MM-DiTs and decouples the entangled text-image attention, inspired by the attention analysis presented in DiTCtrl Cai et al. (2025). This decoupling mechanism, in turn, enables precise text-guided semantic editing without introducing additional computational overhead.

## 2.3 Unique Contributions

Compared to existing methods, our unique contributions include: (1) We incorporate the RK method into the RF sampling process to perform high-order modeling of the differential trajectory, and propose a high-fidelity inversion method that better aligns the inversion and denoising paths. (2) We introduce a decoupled attention mechanism that decouples the entangled text and image attention in MM-DiTs, thereby enabling precise semantic editing in MM-DiT architectures. (3) Extensive experimental results on benchmark datasets demonstrate that our method achieves superior performance on both reconstruction and text-guided semantic editing tasks.

## 3 The Proposed Method

In this section, we first provide a brief overview of the relevant background knowledge and our method, followed by detailed descriptions of the proposed Runge-Kutta solver and DDTA.

### 3.1 Preliminaries and Method Overview

**(1) Preliminaries.** Rectified Flow (RF) Liu et al. (2023) transits the standard Gaussian noise (source) distribution $p_1$ to the real data (target) distribution $p_0$ along a straight path. This transition is modeled by an ordinary differential equation (ODE) over a continuous time interval $t \in [0, 1]$:

$$dZ_t = v\left(Z_t, t\right) dt, \tag{1}$$

where $Z_0 \sim p_0$ denotes the image latent representation sampled from the target distribution, and $Z_1 \sim p_1$ is the noise latent sampled from the source distribution $\mathcal{N}\left(0, \mathrm{I}\right)$. Given an initial state $Z_0$ and a terminal state $Z_1$, the forward process (*i.e.*, adding noise) of RF follows a linear path defined as $Z_t = tZ_1 + (1 - t) Z_0$. This path induces a corresponding ODE: $dZ_t = (Z_1 - Z_0) dt$. Then, the training process employs a diffusion transformer $v_\theta$, parameterized by $\theta$, to approximate the ODE by solving the following least-squares regression objective:

$$\min_\theta \int_0^1 \mathbb{E}\left[\left\|\frac{dZ_t}{dt} - v_\theta\left(Z_t, t\right)\right\|_2^2\right]. \tag{2}$$

In practice, the ODE is discretized and solved using the Euler method for the text-to-image RF models. Specifically, the RF process begins with a noise latent $Z_{t_N} \in \mathcal{N}(0, I)$, and performs denoising over $N$ discrete timesteps $t = \{t_N, \ldots, t_0\}$, progressively refining the latent representation until the final image latent $Z_{t_0}$ is obtained:

$$Z_{t_{i-1}} = Z_{t_i} + (t_{i-1} - t_i) v_\theta (Z_{t_i}, t_i, \mathcal{P}), \tag{3}$$

where $\mathcal{P}$ is the conditional prompt embedding extracted by the T5 text encoder Raffel et al. (2020).

**(2) Method Overview.** We identify two key challenges that lead to the imbalance between fidelity and editability in RF-based image inversion and semantic editing: (1) the low inversion accuracy which hinders the faithfulness between the edited image and the source image, and (2) the entanglement of text and image modalities in MM-DiTs limits precise control over attention features. To address these two tightly coupled challenges, we first incorporate the Runge-Kutta method with RF models and propose the RK Solver to perform high-order approximations of the differential process. Secondly, we propose the DDTA mechanism, which improves attention controllability by disentangling text and image modalities in MM-DiTs, thereby achieving a better balance between fidelity and editability.

## 3.2 RUNGE-KUTTA SOLVER FOR RECTIFIED FLOW MODELS

Despite the impressive image generation performance of the vanilla RF sampler, it suffers from severely degraded fidelity in inversion. RF-Solver Wang et al. (2025) has shown that high-order solvers can partially alleviate this issue. However, it merely presents a second-order method derived from Taylor expansion, without thoroughly exploring the combination of high-order terms, and this ultimately results in suboptimal performance. Prior work Liu et al. (2023) has demonstrated that the outputs of ODEs tend to fall into a smooth manifold. The smoothness of the learned ODE trajectory in RF models arises from the linear interpolation in the forward process, making the system a non-stiff differential equation. Therefore, the well-studied explicit Runge-Kutta (RK) method from numerical analysis becomes a naturally suitable high-order solver for the ODE in RF.

We now present the inversion form of the proposed RK solver. Given a known state at the $(i-1)$-th timestep $Z_{t_{i-1}}$, an $r$-order of explicit RK solver builds a series of intermediate slopes $\{K_1^i, \ldots, K_r^i\}$:

$$K_s^i = v_\theta \left( Z_{t_{i-1}} + \Delta t_i \sum_{j=1}^{s-1} a_{sj} K_j^i, t_{i-1} + c_s \Delta t_i, \mathcal{P} \right), \tag{4}$$

where $\Delta t_i = t_i - t_{i-1}$ denotes the step size of adjacent states. Then, the next state $Z_{t_i}$ can be computed by:

$$Z_{t_i} = Z_{t_{i-1}} + \Delta t_i \sum_{j=1}^{r} b_j K_j^i. \tag{5}$$

Note that the lower triangular matrix $\mathbf{A} = [a_{mn}]$ together with the vectors $\mathbf{b}^T$ and $\mathbf{c}$, constitute a Butcher tableau, *i.e.*,

$$\mathbf{B} = \begin{array}{c|c} \mathbf{c} & \mathbf{A} \\ \hline & \mathbf{b}^T \end{array}. \tag{6}$$

The denoising process of the RK solver has a symmetrical formulation, *i.e.*,

$$K_s^i = v_\theta \left( Z_{t_i} - \Delta t_i \sum_{j=1}^{s-1} a_{sj} K_j^i, t_i - c_s \Delta t_i, \mathcal{P} \right), \tag{7}$$

$$Z_{t_{i-1}} = Z_{t_i} - \Delta t_i \sum_{j=1}^{r} b_j K_j^i. \tag{8}$$

Empirically, the RK solver should adopt the same order for both the inversion and denoising processes. The complete inversion process with our RK solver is provided in pseudocode in the Appendix.

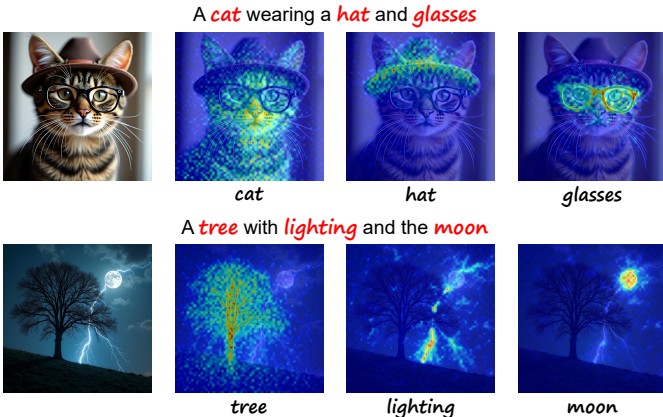

Figure 2: Visualization of the response maps corresponding to the decoupled cross-attention components. We aggregate cross-attention maps across all DiT blocks during the sampling process to show the spatial correlation between image layout and prompt words. Details see in Appendix B.3.

Noting that the existing inversion methods, RF-Solver and FireFlow, can be regarded as two specific variants of our RK solver. RF-Solver uses the Taylor expansion to approximate the velocity prediction, while FireFlow uses the midpoint method. Thus, their Butcher tableaus are given by:

$$\mathbf{B}_{\text{RF-Solver}} = \begin{array}{c|cc} 0 & 0 & 0 \\ \frac{1}{2} & \frac{1}{2} & 0 \\ \hline & \frac{3}{4} & \frac{1}{4} \end{array} \;, \quad \mathbf{B}_{\text{FireFlow}} = \begin{array}{c|cc} 0 & 0 & 0 \\ \frac{1}{2} & \frac{1}{2} & 0 \\ \hline & 0 & 1 \end{array} \;. \tag{9}$$

In addition, FireFlow achieves runtime acceleration by reusing the midpoint velocity from the previous step to approximate the current velocity, *i.e.*, $K_1^i \approx K_2^{i-1}$.

### 3.3 DECOUPLED DIFFUSION TRANSFORMER ATTENTION FOR SEMANTIC IMAGE EDITING

Text-guided semantic editing attracts the most attention due to its flexibility. Representative methods (*e.g.*, P2P Hertz et al. (2023)) focus on manipulating attentions in the editing branch by leveraging the preserved attentions from the inversion branch. The effectiveness of these methods stems from the separate design of self-attention and cross-attention mechanisms in UNet-based diffusion models. However, the MM-DiT-based diffusion models process text and image information jointly within a unified transformer framework, making it difficult to transfer those effective methods to DiT-based architectures. Specifically, the MM-DiT architecture consists of two types of transformer blocks, *i.e.*, multi-stream and single-stream DiT blocks. In the multi-stream DiT block, text and image attention features are first extracted separately:

$$\mathcal{F}_C^l = W_{\mathcal{F}_C}^l \left( h_C^l \right), \quad \mathcal{F}_I^l \left( t_i \right) = W_{\mathcal{F}_I}^l \left( h_I^l \left( t_i \right) \right) . \tag{10}$$

Here, $l$ denotes the layer index of the DiT block. $W(\cdot)$ represents the pre-trained attention projection in the transformer. $\mathcal{F}_C = \{Q_C, K_C, V_C\}$ is the attention feature corresponding to the conditional hidden state related to the textual prompt, while $\mathcal{F}_I(t_i) = \{Q_I(t_i), K_I(t_i), V_I(t_i)\}$ correspond to the attention feature derived from the hidden state associated with the image latent at timestep $t_i$. Then, attention features are concatenated $\mathcal{F}^l = \mathcal{F}_C^l \oplus \mathcal{F}_I^l(t_i)$, followed by the attention computation:

$$\text{Attention}(Q, K, V) = \text{softmax}\left(\frac{QK^T}{\sqrt{d}}\right) \cdot V . \tag{11}$$

In the single-stream DiT block, text and image hidden states are concatenated before attention feature extraction, *i.e.*,

$$\mathcal{F}^l(t_i) = W_{\mathcal{F}}^l \left( h_C^l \oplus h_I^l(t_i) \right) . \tag{12}$$

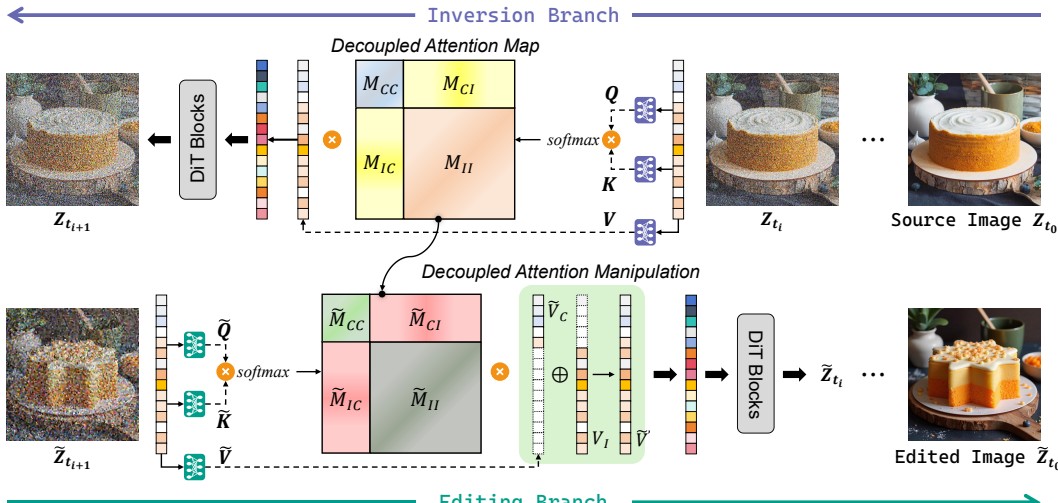

Prompt: a ~~round~~ *star* cake with orange frosting on a wooden plate

Figure 3: Overview of Decoupled Diffusion Transformer Attention (DDTA)

According to the observation of the internal structure, the attention map of the DiT block can be decoupled into four regions based on the dimension of hidden states:

$$M = \text{softmax}\left(\frac{QK^T}{\sqrt{d}}\right) = \left[\begin{array}{c|c} M_{CC} & M_{CI} \\ \hline M_{IC} & M_{II} \end{array}\right] , \tag{13}$$

where $M_{CC}$ and $M_{II}$ correspond to the self-attention maps of the condition and image, while $M_{CI}$ and $M_{IC}$ represent the cross-attention maps between condition and image. As shown in Fig. 2, the decoupled cross-attention maps reveal strong spatial correlations between image layout and prompt words, demonstrating the effectiveness of our attention decoupling method. In addition, the value feature can also be decoupled into two regions according to the dimension: $V = [V_C|V_I]$.

Therefore, the proposed attention decoupling mechanism facilitates effective text-guided semantic editing via fine-grained attention manipulation. Specifically, as illustrated in Fig. 3, attentions in the editing branch $B_{\text{edit}} = \left\{\widetilde{M}_{CC}, \widetilde{M}_{CI}, \widetilde{M}_{IC}, \widetilde{M}_{II}, \widetilde{V}\right\}$ are modified based on the preserved attentions in the inversion branch $B_{\text{inv}} = \{M_{CC}, M_{CI}, M_{IC}, M_{II}, V\}$. We consider two types of operations, *i.e.*,

$$\text{Replacement: } \widetilde{M}' = M, \quad \text{Mean: } \widetilde{M}' = \left(M + \widetilde{M}\right)/2 . \tag{14}$$

Both operations improve the faithfulness of the edited image, with the replacement strategy yielding a greater fidelity gain than the mean operation. This indicates that preserving original attention features is more effective in maintaining content consistency. Moreover, incorporating a larger proportion of attention features into the manipulation enhances fidelity to the source image but reduces editability. These findings highlight the inherent trade-off between fidelity and editability, which must be carefully balanced in image editing tasks. For general semantic editing purposes, applying the replacement operation to cross-attention maps $\left\{\widetilde{M}_{CI}, \widetilde{M}_{IC}\right\}$ and the mean operation to the image region of value attention feature $\widetilde{V}_I$ in single-stream DiT blocks is typically sufficient. Furthermore, users can flexibly customize both the manipulation type and the number of blocks or sampling steps to achieve the most satisfactory result for a given image.

## 4 EXPERIMENTAL RESULTS

In this section, we conduct comprehensive evaluations of the proposed method and provide detailed ablation studies to further understand its performance. Detailed experimental settings are presented in Appendix B. More results are provided in Appendix C.

Table 1: Comparison results on the reconstruction task. The best result for each metric is highlighted in bold, and the second-best is underlined (excluding the VQAE method).

| Method | PSNR ↑ | SSIM ↑ | LPIPS ↓ |
|---|---|---|---|
| VQAE (upper bound) | 32.95 | 0.9347 | 0.0121 |
| Vanilla RF | 17.46 | 0.5952 | 0.4282 |
| RF Inversion | 22.14 | 0.6540 | 0.1388 |
| RF-Solver | 22.20 | 0.7778 | 0.1890 |
| Fireflow | 23.29 | 0.8006 | 0.1639 |
| Ours ($r = 2$) | 24.00 | 0.8124 | 0.1534 |
| Ours ($r = 3$) | 23.98 | 0.8131 | 0.1497 |
| Ours ($r = 4$) | **25.68** | **0.8364** | **0.1241** |

Table 2: Comparison results on the text-guided semantic editing task. The best result for each metric is highlighted in bold, and the second-best is underlined.

| Method | Baseline | Structure Distance↓ | Unedited Fidelity PSNR ↑ | SSIM ↑ | CLIP Similariy Whole ↑ | Edited ↑ | Steps |
|---|---|---|---|---|---|---|---|
| P2P | SD | 0.0699 | 17.84 | 0.7141 | 25.18 | 22.35 | 50 |
| MasaCtrl | SD | 0.0277 | 22.31 | 0.8041 | 23.99 | 21.15 | 50 |
| PnP | SD | 0.0273 | 22.32 | 0.7958 | **25.42** | 22.52 | 50 |
| RF Inversion | FLUX | 0.0446 | 20.31 | 0.7014 | 25.07 | 22.36 | 28 |
| RF-Solver | FLUX | 0.0297 | 22.27 | 0.7938 | 24.61 | 21.87 | 25 |
| FireFlow | FLUX | 0.0264 | 23.30 | 0.8302 | 24.53 | 21.65 | 8 |
| Ours ($r = 2$) | FLUX | 0.0288 | 23.29 | 0.8296 | 25.30 | **22.54** | 8 |
| Ours ($r = 3$) | FLUX | 0.0284 | 23.51 | 0.8339 | 25.30 | 22.50 | 8 |
| Ours ($r = 4$) | FLUX | **0.0259** | **24.24** | **0.8535** | 24.67 | 21.95 | 8 |
| Ours ($r = 4$) | FLUX | 0.0271 | 23.76 | 0.8431 | 25.26 | 22.48 | 5 |

## 4.1 Image Reconstruction Task

We compare our proposed RK solver against VQAE, vanilla RF, RF inversion, RF-Solver, and Fire-Flow. The evaluations are conducted on the first 1,000 images from the Densely Captioned Images (DCI) dataset Urbanek et al. (2024), using the same experimental setting as in FireFlow's original literature Deng et al. (2025). The VQAE method represents the upper bound of reconstruction performance, as it directly decodes the image latent $Z_{t_0}$ obtained from the encoder. The results are presented in Tab. 1, demonstrating that our method outperforms all existing approaches across all evaluation metrics. We report the results based on the best-performing configurations, using Heun's second-order, Kutta's third-order, and the 3/8-rule fourth-order Kutta (1901) variants, whose corresponding Butcher tableaus are as follows:

$$
\mathbf{B}_{r=2} = \begin{array}{c|cc} 0 & 0 & 0 \\ 1 & 1 & 0 \\ \hline & \frac{1}{2} & \frac{1}{2} \end{array}, \quad
\mathbf{B}_{r=3} = \begin{array}{c|ccc} 0 & 0 & 0 & 0 \\ \frac{1}{2} & \frac{1}{2} & 0 & 0 \\ 1 & -1 & 2 & 0 \\ \hline & \frac{1}{6} & \frac{2}{3} & \frac{1}{6} \end{array}, \quad
\mathbf{B}_{r=4} = \begin{array}{c|cccc} 0 & 0 & 0 & 0 & 0 \\ \frac{1}{3} & \frac{1}{3} & 0 & 0 & 0 \\ \frac{2}{3} & -\frac{1}{3} & 1 & 0 & 0 \\ 1 & 1 & -1 & 1 & 0 \\ \hline & \frac{1}{8} & \frac{3}{8} & \frac{3}{8} & \frac{1}{8} \end{array}.
$$

The third-order variant slightly outperforms the second-order one, while our fourth-order variant achieves the best reconstruction performance, surpassing FireFlow by 10.3%, 4.5%, and 24.3% in PSNR, SSIM, and LPIPS, respectively.

## 4.2 Text-Guided Semantic Editing Task

**Quantitative Results.** We conduct a comprehensive quantitative comparison on the PIE-bench dataset Ju et al. (2024) across various methods, including both DDIM-based and RF-based methods,

Table 3: User study of the text-guided semantic editing task on the PIE-Bench dataset.

| Method | Qwen-VL-Max | Hunyuan-T1 | Doubao-1.5 |
|---|---|---|---|
| P2P | 4.01 | 10.74 | 6.59 |
| MasaCtrl | 14.59 | 9.31 | 7.74 |
| PnP | 15.74 | 14.18 | 17.34 |
| RF Inversion | 12.59 | 10.89 | 7.88 |
| RF-Solver | 11.59 | 13.75 | 6.03 |
| FireFlow | 8.44 | 15.75 | 10.32 |
| Ours | **33.04** | **25.36** | **44.13** |

Table 4: Ablation study on the trade-off between fidelity and editability.

| RK Solver | DDTA | Structure Distance↓ | Unedited Fidelity | | CLIP Similariy | |
|---|---|---|---|---|---|---|
| | | | PSNR ↑ | SSIM ↑ | Whole ↑ | Edited ↑ |
| ✗ | ✗ | 0.0264 | 23.30 | 0.8302 | 24.53 | 21.65 |
| ✓ | ✗ | **0.0133** | **28.42** | **0.8918** | 23.74 | 21.00 |
| ✗ | ✓ | 0.0358 | 22.18 | 0.8177 | **25.73** | **22.99** |
| ✓ | ✓ | 0.0284 | 23.51 | 0.8339 | 25.30 | 22.50 |

using Stable Diffusion v1.5 and FLUX.1-dev as their respective baselines. Noting that the results of our method presented in this section are achieved through the combination of the proposed RK Solver and DDTA. Quantitative results shown in Tab. 2 support the following three conclusions: (1) Our method outperforms all baselines in content consistency, with our fourth-order variant achieving the highest PSNR, SSIM, and structure distance. (2) Our method demonstrates competitive editability (closely trailing the best result) while maintaining substantially higher fidelity, highlighting a more favorable trade-off between fidelity and editability. (3) Our method achieves the best overall performance with significantly fewer sampling steps, indicating the superior efficiency of our method.

**Qualitative Results.** As shown in Fig. 4, we present qualitative results demonstrating the effectiveness of our method across diverse editing types, including both object and attribute manipulations. While minor unintended background changes may occur, our method consistently outperforms existing baselines in terms of semantic alignment with target prompts and structural consistency with source images, highlighting its robustness and versatility in the text-guided semantic editing task.

**User Study.** To further evaluate the effectiveness of our proposed method, we employ Multimodal Large Language Models (MLLMs) to assess the quality of edited images based on both editing performance and consistency with the source image. To ensure the reliability of the evaluation, we select three state-of-the-art MLLMs as independent judges: Qwen-VL-Max, Hunyuan-T1, and Doubao-1.5. This evaluation is performed on the entire PIE-Bench dataset, where for each image, the MLLMs are tasked with selecting the best-edited result among all compared methods. In this study, we adopt the fourth-order variant with 5 sampling steps for comparison against other baselines. We report the proportion of selections for each method across the dataset. As shown in Tab. 3, our proposed method is selected significantly more often than all comparisons, demonstrating its superior editing quality and faithfulness.

## 4.3 ABLATION STUDIES

To assess the contribution of each component in our framework, we perform an ablation study on the PIE-Bench dataset. We use FireFlow as the baseline and compare it against our third-order variant under a fixed setting of 8 sampling steps. As shown in Tab. 4, our full framework, which integrates the proposed RK Solver and DDTA, achieves the best balance between fidelity and editability. Specifically, replacing the baseline sampler with the RK Solver significantly improves content consistency, though the alignment with the editing prompt remains limited. In contrast, substituting the editing module with DDTA leads to substantial gains in editability, but at the cost of significantly reduced fidelity. Additional ablation studies are provided in the Appendix, including the selection of

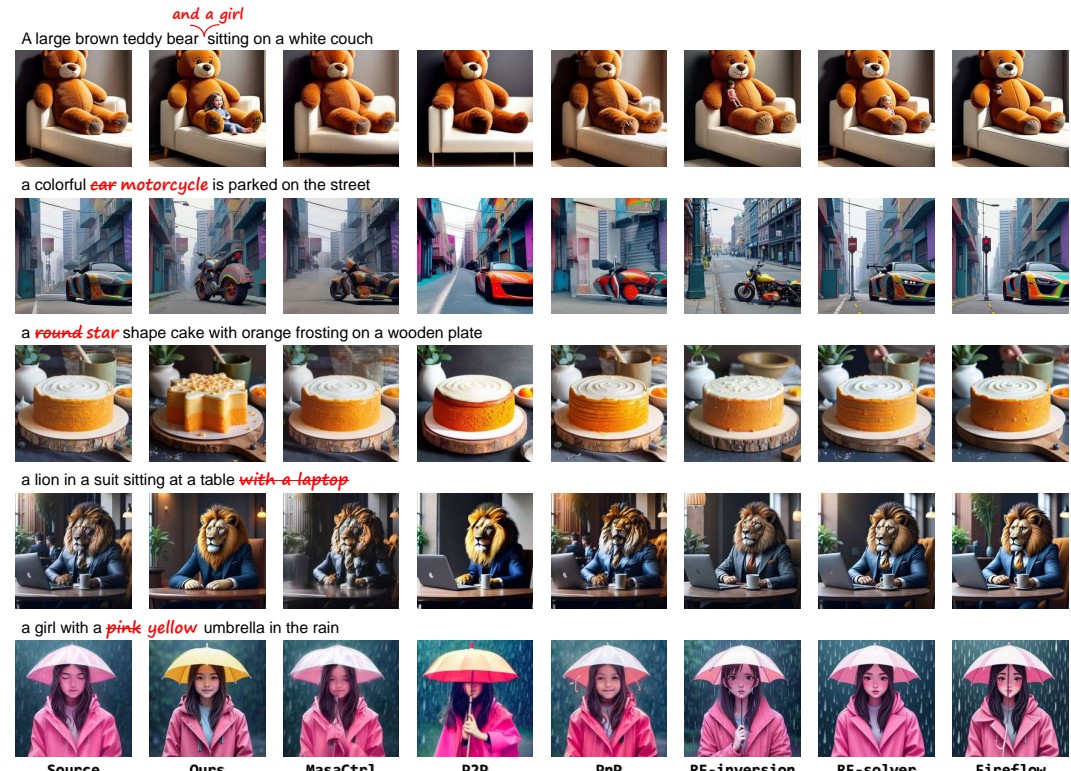

Figure 4: Qualitative results on the text-guided semantic editing task.

the Butcher tableau (C.1), the sampling steps and Number of Function Evaluations (C.2), the contribution of each decoupled attention component (C.3), and the effectiveness of different attention manipulation strategies (C.4).

## 5 CONCLUSION

In this work, we address two critical limitations of applying RF models to real-world image editing: (1) the difficulty of accurate inversion, and (2) the lack of semantic controllability stemming from entangled multimodal attention. To overcome the first challenge, we propose RK Solver, a high-order inversion technique inspired by the well-studied Runge-Kutta method from numerical analysis. To address the second challenge, we introduce DDTA, a novel attention mechanism that decouples text and image modalities in MM-DiTs. Extensive experimental results on image reconstruction and text-guided editing tasks demonstrate the effectiveness of our approach, which achieves the SoTA performance in terms of fidelity and editability. Regarding the societal impact of this work, our method not only enhances the practical applicability of RF models in real-world generative tasks but also provides new insights into controllable diffusion-based generation. These advancements have the potential to benefit various domains, including the creative industries, digital art, education, and accessibility. Although the proposed RK solver significantly improves fidelity, its high-order modeling introduces additional computational overhead. Additionally, preserving the decoupled attention maps incurs notable memory consumption. These limitations suggest promising directions for future work, including the development of high-order solvers with low computational overhead and the design of efficient attention-preserving mechanisms, which could further improve the practicality of RF models.

## 6 ETHICS STATEMENT

The proposed editing framework presents both positive and negative societal impacts. On the positive side, it enables relatively fine-grained and flexible editing of real-world images through simple modifications to textual descriptions, which may benefit applications in creative industries, digital art, and education. On the negative side, the method could be misused by malicious actors to generate inappropriate or offensive content. In particular, the high-order RK solver may exacerbate the inherent risks associated with the underlying generative models.

## 7 REPRODUCIBILITY STATEMENT

We are committed to ensuring the full reproducibility of the work presented in this paper, and have made systematic efforts to provide comprehensive supporting materials and clear references to critical details. Specifically, we provide a detailed theoretical analysis in Appendix A and complete implementation details in Appendix B.3 to eliminate ambiguity in practical deployment. In addition, the anonymous source code repository corresponding to this paper is available at `https://anonymous.4open.science/r/653A143A5BF8`.

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

## A  APPENDIX: THEORETICAL ANALYSIS

In practice, $v_\theta$ is predicted by a neural network without a rigorous mathematical explicit expression. It may be unable to be exactly equal to the original required $v$ for various reasons, including the performance of the neural network and the noise. This error could accumulate when finding the solution to the ODE by iterative methods. Therefore, understanding the bounds of this error is crucial for assessing and improving the denoising algorithm of flow matching. In this section, we will prove that this error has an upper bound as

$$|\widetilde{Z}_{t_0} - Z_{t_0}| \le e^\Lambda |\delta_0| + \frac{e^\Lambda - 1}{\Lambda} \max_{1 \le i \le N} |\delta_i|, \tag{15}$$

where $\widetilde{Z}_{t_0}$ is the ODE solution with perturbation, $Z_{t_0}$ is the ODE solution by iterative method without any perturbation ideally, $\delta_i (i = 0, 1, 2, \ldots, N)$ is the perturbation at step $i$, and $\Lambda = (41/24)L$ ($L$ is Lipschitz constant).

*Proof.* The process of solving ODE is determining $Z_{t_0}$ based on the given $Z_{t_N}$. This can be achieved using the general Runge-Kutta method, which takes the form:

$$Z_{t_{i-1}} = Z_{t_i} + h\Phi_i(Z_{t_i}) \ (i = 1, 2, 3, \ldots, N), \tag{16}$$

where $h$ is the hyperparameter associated with the time step configuration, and $\Phi_i(Z_{t_i})$ denotes the incremental function of the Runge-Kutta method at step $i$. Since the time step in RF is uniform ($\Delta t_i = \Delta t_j$ for $i \ne j$)), it can be simplified that $h = \Delta t_i (i = 0, 1, 2, \ldots, N)$. And the $\Phi_i(Z_{t_i})$ is defined as:

$$\Phi_i(Z_{t_i}) = \sum_{j=1}^{r} b_j K_j^i(Z_{t_i}), \text{ where } K_s^i(Z_{t_i}) = v_\theta \left( Z_{t_i} - \Delta t_i \sum_{j=1}^{s-1} a_{sj} K_j^i, t_i - c_s \Delta t_i, \mathcal{P} \right). \tag{17}$$

In conjunction with Equation (16), the perturbed process can be represented as:

$$\widetilde{Z}_{t_{i-1}} = \widetilde{Z}_{t_i} + h \left[ \Phi_i(\widetilde{Z}_{t_i}) + \delta_i \right] \ (i = 1, 2, 3, \ldots, N), \ \widetilde{Z}_{t_N} = Z_{t_N} + \delta_N. \tag{18}$$

According to Runge Runge (1895), it is shown that for the 4-th order RF solver in this work:

$$|\Phi_i(y) - \Phi_i(x)| \le \left[ 1 + \frac{Lh}{2!} + \frac{(Lh)^2}{3!} + \frac{(Lh)^4}{4!} \right] L|y - x|, \tag{19}$$

where $L$ can be any Lipschitz constant of $v$. And from Equation (19), it can be obtained that

$$|\Phi_i(y) - \Phi_i(x)| \le \Lambda |y - x|, \text{ for } 0 < h \le h_0. \tag{20}$$

Take the $h_0 = 1/L$, it can be obtained that $\Lambda = (41/24)L$.

By substracting the ralations in Equation (16) from the corresponding ones in Equation (18) and by using the Equation (19), it can be obtained that

$$|\widetilde{Z}_{t_{i-1}} - Z_{t_{i-1}}| \le (1 + h\Lambda)|\widetilde{Z}_{t_i} - Z_{t_i}| + h|\delta_{i-1}| \ (i = 1, 2, 3, \ldots, N). \tag{21}$$

As $(1 + h\Lambda)^n \le e^{nh\Lambda}$, it can be iteratively obtained:

$$|\widetilde{Z}_{t_0} - Z_{t_0}| \le e^{\Lambda T} |\delta_0| + \frac{e^{\Lambda T} - 1}{\Lambda} \max_{1 \le i \le N} |\delta_i|. \tag{22}$$

## B  APPENDIX: EXPERIMENTAL SETTINGS

In this section, we present detailed experimental settings of this paper.

### B.1  BASELINES

We adopt FLUX.1-dev with the vanilla RF Euler sampler as the baseline for all tasks. For the reconstruction task, we compare SoTA inversion approaches designed for RF models, such as RF inversion Rout et al. (2025), RF-Solver Wang et al. (2025), and FireFlow Deng et al. (2025). For the editing task, we include both RF-based methods and DDIM-based approaches for comparison. Here, the DDIM-based methods include P2P Hertz et al. (2023), MasaCtrl Cao et al. (2023), and PnP Tumanyan et al. (2023).

---

**Algorithm 1:** Semantic Editing Using RK Solver and DDTA

---

**Input:** Source image $Z_{t_0}$, Source prompt $\mathcal{P}_s$, Target prompt $\mathcal{P}_t$, $r$-order Butcher tableau $\mathbf{B}_r$,
         Sampling steps $N$, Index list for performing DDTA $\mathbf{D}_{\text{list}}$

**Output:** Target image $\widetilde{Z}_{t_0}$

---

    `// Inversion Stage`

1   $c \leftarrow N \times r$

2   **for** $i \leftarrow 1$ **to** $N$ **do**

3      $\Delta t_i \leftarrow t_i - t_{i-1}$

4      $Z_{t_i} \leftarrow Z_{t_{i-1}}$

5      **for** $s \leftarrow 1$ **to** $r$ **do**

6         **if** $c$ *in* $\mathbf{D}_{list}$ **then**

7            $K_s^i \leftarrow \text{DDTA}_{\text{save}} \left( Z_{t_{i-1}} + \Delta t_i \sum_{j=1}^{s-1} a_{sj} K_j^i, t_{i-1} + c_s \Delta t_i, \mathcal{P}_s \right)$

8         **else**

9            $K_s^i \leftarrow v_\theta \left( Z_{t_{i-1}} + \Delta t_i \sum_{j=1}^{s-1} a_{sj} K_j^i, t_{i-1} + c_s \Delta t_i, \mathcal{P}_s \right)$

10       $Z_{t_i} \leftarrow Z_{t_i} + b_s \Delta t_i K_s^i$

11       $c \leftarrow c - 1$

    `// Editing Stage`

12   $\widetilde{Z}_{t_N} \leftarrow Z_{t_N}$

13   $c \leftarrow 1$

14   **for** $i \leftarrow N$ **to** $1$ **do**

15      $\Delta t_i \leftarrow t_i - t_{i-1}$

16      $\widetilde{Z}_{t_{i-1}} \leftarrow \widetilde{Z}_{t_i}$

17      **for** $s \leftarrow 1$ **to** $r$ **do**

18         **if** $c$ *in* $\mathbf{D}_{list}$ **then**

19            $\widetilde{K}_s^i \leftarrow \text{DDTA}_{\text{manipulate}} \left( \widetilde{Z}_{t_i} - \Delta t_i \sum_{j=1}^{s-1} a_{sj} \widetilde{K}_j^i, t_i - c_s \Delta t_i, \mathcal{P}_t \right)$

20         **else**

21            $\widetilde{K}_s^i \leftarrow v_\theta \left( \widetilde{Z}_{t_i} - \Delta t_i \sum_{j=1}^{s-1} a_{sj} \widetilde{K}_j^i, t_i - c_s \Delta t_i, \mathcal{P}_t \right)$

22       $\widetilde{Z}_{t_{i-1}} \leftarrow \widetilde{Z}_{t_i} - b_s \Delta t_i \widetilde{K}_s^i$

23       $c \leftarrow c + 1$

24   **return** $\widetilde{Z}_{t_0}$

---

### B.2 DATASETS AND EVALUATION METRICS

We evaluate the proposed method on two tasks: image reconstruction and text-guided semantic editing. To comprehensively assess the reconstruction performance of our RK solver, we report results on the first 1,000 images from the Densely Captioned Images (DCI) dataset Urbanek et al. (2024), using Peak Signal-to-Noise Ratio (PSNR), Structural Similarity Index Measure (SSIM) Wang et al. (2004), and Learned Perceptual Image Patch Similarity (LPIPS) Zhang et al. (2018) as evaluation metrics. For the editing task, we assess our framework on the PIE-Bench dataset Ju et al. (2024). We use CLIP Radford et al. (2021) to measure the alignment between the edited image and the guiding text. To evaluate the fidelity of non-edited regions, we further report PSNR, SSIM, and structural distance Ju et al. (2024).

### B.3 IMPLEMENTATION DETAILS

All methods are implemented by PyTorch and the Diffusers library, with all reported results based on our re-implementation. All experiments are conducted on a single NVIDIA A100-80G SXM GPU, unless otherwise specified.

---

**Algorithm 2:** Visualization of Word-Pixel Response Maps for MM-DiT Architectures

---

**Input:** Input caption with $k$ words $\mathcal{C} = \{w_1, \ldots, w_k\}$, Target word index $G$
**Output:** Word-pixel response maps $R = \{R_g | g \in G\}$

1   $z_{t_N} \leftarrow \mathcal{N}(0, \mathbf{I})$   // sample initial latent
2   $\mathcal{P} = \text{T5}(\mathcal{C})$   // compute prompt embedding
3   $A_{\text{cache}} \leftarrow$ None   // initialize cached decoupled attention maps
4   $R \leftarrow []$
5   **for** $i \leftarrow N$ **to** 1 **do**
6     $\lfloor$   $z_{t_{i-1}} \leftarrow z_{t_i} + (t_{i-1} - t_i) \cdot v_\theta^{\text{cache}}(z_{t_i}, t_i, \mathcal{P}, A_{\text{cache}})$
7   $A_{\text{cache}} \leftarrow A_{\text{cache}}/N$
8   **for** $g \in G$ **do**
9     $A_g \leftarrow A_{\text{cache}}[:, g]$ // extract target attention maps
10    $A_g \leftarrow \text{resize}(A_g, H \times W)$ // resize to image resolution
11    $R.\text{append}(A_g)$

12   **Function** $v_\theta^{\text{cache}}$ ($z_{t_i}$, $t_i$, $\mathcal{P}$, $A_{\text{cache}}$)**:**
    // multi-stream DiT blocks
13    $h_C \leftarrow \text{AdaLayerNorm}(\mathcal{P}, t_i)$
14    $h_I \leftarrow \text{AdaLayerNorm}(z_{t_i}, t_i)$
15    **for** $l \leftarrow 1$ **to** $L_{\text{multi}}$ **do**
16      $Q_C, K_C, V_C \leftarrow W_{Q_C}^l(h_C), W_{K_C}^l(h_C), W_{V_C}^l(h_C)$
17      $Q_I, K_I, V_I \leftarrow W_{Q_I}^l(h_I), W_{K_I}^l(h_I), W_{V_I}^l(h_I)$
18      $Q, K, V \leftarrow Q_C \oplus Q_I, K_C \oplus K_I, V_C \oplus V_I$
19      $M\{M_{CC}, M_{CI}, M_{IC}, M_{II}\} \leftarrow \text{softmax}(\frac{QK^T}{\sqrt{\dim}})$
20      **if** $A_{\text{cache}}$ *is None* **then**
21       $\lfloor$   $A_{\text{cache}} \leftarrow M_{IC} + M_{CI}^T$
22      **else**
23       $\lfloor$   $A_{\text{cache}} \leftarrow A_{\text{cache}} + M_{IC} + M_{CI}^T$
24      $\{h_C, h_I\} \leftarrow M \cdot V$
    // single-stream DiT blocks
25    $h \leftarrow h_{\mathcal{C}} \oplus h_{\mathcal{I}}$
26    **for** $l \leftarrow$ **to** $L_{\text{single}}$ **do**
27      $Q, K, V \leftarrow W_Q^l(h), W_K^l(h), W_V^l(h)$
28      $M\{M_{CC}, M_{CI}, M_{IC}, M_{II}\} \leftarrow \text{softmax}(\frac{QK^T}{\sqrt{\dim}})$
29      $A_{\text{cache}} \leftarrow A_{\text{cache}} + M_{IC} + M_{CI}^T$
30      $h \leftarrow M \cdot V$
31    $v_{t_i} \leftarrow \text{PostProcess}(h)$
32    $A_{\text{cache}} \leftarrow A_{\text{cache}}/(L_{\text{multi}} + L_{\text{single}})$
33    **return** $v_{t_i}$
34   **return** $R$

---

**Image Reconstruction Task.** Images from the DCI dataset are center-cropped to square format and resized to $1024 \times 1024$, using the short version of the provided captions. The guidance scale of each method is set to 1.0. The sampling steps of each method are set to 30, except for VQAE. The hyperparameter settings for RF inversion follow the configuration in the literature Rout et al. (2025), with controller guidance $\gamma = 0.5$, $\eta = 1.0$, starting time $s = 8$, and stopping time $\tau = 25$.

**Text-Guided Semantic Editing Task.** Stable Diffusion v1.5 is adopted as the baseline model for all DDIM-based methods. For DDIM-based methods, the guidance scales for the inversion and editing branches are set to 1.0 and 7.5, respectively. For RF-based methods, these scales are set to 1.0 and 3.0, respectively. For RF inversion, we report the best editing results on the PIE-Bench dataset using controller guidance parameters $\gamma = 0.5$, $\eta = 0.9$, with a starting time $s = 0$ and stopping time

Table 5: Ablation study on Butcher tableaux across second- to fourth-order solvers. Results in bold denote the best performance for each solver order.

| Variant | Order | PSNR ↑ | SSIM ↑ | LPIPS ↓ |
|---------|-------|--------|--------|---------|
| Midpoint | 2 | 22.17 | 0.7766 | 0.1903 |
| Heun | 2 | **24.00** | **0.8124** | **0.1534** |
| Ralston | 2 | 23.73 | 0.8096 | 0.1556 |
| Kutta | 3 | **23.98** | **0.8131** | **0.1497** |
| Heun | 3 | 22.48 | 0.7780 | 0.1936 |
| Ralston | 3 | 21.46 | 0.7611 | 0.2061 |
| Houwen | 3 | 21.11 | 0.7488 | 0.2270 |
| SSPRK3 | 3 | 20.52 | 0.7364 | 0.2373 |
| Classic | 4 | 24.46 | 0.8197 | 0.1425 |
| 3/8-rule | 4 | **25.68** | **0.8364** | **0.1241** |
| Ralston | 4 | 20.16 | 0.7258 | 0.2564 |

$\tau = 6$. To minimize the impact of image resolution, we resize the images from the PIE-Bench dataset Ju et al. (2024) to match the resolution used in the corresponding pre-trained baselines, *i.e.*, $512 \times 512$ for SD and $1024 \times 1024$ for FLUX. The configuration of our DDTA follows the general setup described in Sec. 3.3, where attention manipulations are applied only to the single-stream DiT blocks at the first sampling step. Specifically, text-guided semantic editing is performed using Algorithm 1, with $\mathbf{D}_{\text{list}} = [1]$. In all single-stream DiT blocks, we replace the cross-attention maps $\{M_{CI}, M_{IC}\}$ and apply a mean operation to the value feature $V_I$.

**Visualization of Word-Pixel Response Maps.** DAAM Tang et al. (2023) is an attention visualization technique originally designed for UNet-based models, and thus cannot be directly applied to MM-DiT architectures. Inspired by DAAM, we visualize word-pixel response maps to validate the correctness of our proposed DDTA. As shown in Fig. 2, the visualization provides direct evidence that DDTA effectively decouples multimodal attention. The implementation details of the visualization are outlined in Algorithm 2.

## C    APPENDIX: ADDITIONAL EXPERIMENTAL RESULTS

In this section, we provide additional experimental results to further demonstrate the effectiveness of the proposed method.

### C.1    ABLATION STUDY ON BUTCHER TABLEAU

The Runge-Kutta method comprises a family of numerical solvers, each defined by a specific Butcher tableau. We report the results obtained using various Butcher tableaux, ranging from second-order to fourth-order solvers. For the second-order methods, we include the midpoint, Heun's, and Ralston's Ralston (1962) methods, which are given by:

$$
\mathbf{B}_{\text{midpoint}}^{(2)} = \begin{array}{c|cc} 0 & 0 & 0 \\ \frac{1}{2} & \frac{1}{2} & 0 \\ \hline & 0 & 1 \end{array}, \quad
\mathbf{B}_{\text{Heun}}^{(2)} = \begin{array}{c|cc} 0 & 0 & 0 \\ 1 & 1 & 0 \\ \hline & \frac{1}{2} & \frac{1}{2} \end{array}, \quad
\mathbf{B}_{\text{Ralston}}^{(2)} = \begin{array}{c|cc} 0 & 0 & 0 \\ \frac{2}{3} & \frac{2}{3} & 0 \\ \hline & \frac{1}{4} & \frac{3}{4} \end{array}.
$$

For the third-order methods, we include the Kutta's Kutta (1901), Heun's, Ralston's Ralston (1962), Van der Houwen's der Houwen P J. (1972), and Strong Stability Preserving Runge-Kutta (SSPRK3) methods, each defined as follows:

$$
\mathbf{B}_{\text{Kutta}}^{(3)} = \begin{array}{c|ccc} 0 & 0 & 0 & 0 \\ \frac{1}{2} & \frac{1}{2} & 0 & 0 \\ 1 & -1 & 2 & 0 \\ \hline & \frac{1}{6} & \frac{2}{3} & \frac{1}{6} \end{array}, \quad
\mathbf{B}_{\text{Heun}}^{(3)} = \begin{array}{c|ccc} 0 & 0 & 0 & 0 \\ \frac{1}{3} & \frac{1}{3} & 0 & 0 \\ \frac{2}{3} & 0 & \frac{2}{3} & 0 \\ \hline & \frac{1}{4} & 0 & \frac{3}{4} \end{array}, \quad
\mathbf{B}_{\text{Ralston}}^{(3)} = \begin{array}{c|ccc} 0 & 0 & 0 & 0 \\ \frac{1}{2} & \frac{1}{2} & 0 & 0 \\ \frac{3}{4} & 0 & \frac{3}{4} & 0 \\ \hline & \frac{2}{9} & \frac{1}{3} & \frac{4}{9} \end{array},
$$

Table 6: Ablation study on sampling steps and NFEs.

| Method | Steps | NFEs | PSNR ↑ | SSIM ↑ | LPIPS ↓ |
|---|---|---|---|---|---|
| Vanilla RF | 30 | 60 | 17.46 | 0.5952 | 0.4282 |
| Vanilla RF | 60 | 120 | 19.03 | 0.6758 | 0.3092 |
| Vanilla RF | 90 | 180 | 19.53 | 0.6969 | 0.2841 |
| Vanilla RF | 120 | 240 | 17.70 | 0.6386 | 0.3539 |
| RF-Solver | 15 | 60 | 19.44 | 0.7157 | 0.2477 |
| RF-Solver | 30 | 120 | 22.20 | 0.7778 | 0.1890 |
| RF-Solver | 60 | 240 | 22.25 | 0.7655 | 0.2106 |
| FireFlow | 30 | 62 | 23.29 | 0.8006 | 0.1639 |
| FireFlow | 60 | 122 | 23.15 | 0.7861 | 0.1873 |
| FireFlow | 90 | 182 | 24.40 | 0.8146 | 0.1496 |
| FireFlow | 120 | 242 | 18.60 | 0.6543 | 0.3311 |
| Ours ($r = 2$) | 15 | 60 | 20.66 | 0.7393 | 0.2407 |
| Ours ($r = 2$) | 30 | 120 | 24.00 | 0.8124 | 0.1534 |
| Ours ($r = 2$) | 60 | 240 | 26.89 | 0.8607 | 0.0974 |
| Ours ($r = 3$) | 30 | 180 | 23.98 | 0.8131 | 0.1497 |
| Ours ($r = 3$) | 40 | 240 | 25.14 | 0.8274 | 0.1349 |
| Ours ($r = 4$) | 15 | 120 | 22.28 | 0.7699 | 0.1993 |
| Ours ($r = 4$) | 30 | 240 | 25.68 | 0.8364 | 0.1241 |

$$
\mathbf{B}^{(3)}_{\text{Houwen}} = \begin{array}{c|ccc}
0 & 0 & 0 & 0 \\
\frac{8}{15} & \frac{8}{15} & 0 & 0 \\
\frac{2}{3} & \frac{1}{4} & \frac{5}{12} & 0 \\
\hline
& \frac{1}{4} & 0 & \frac{3}{4}
\end{array}, \quad
\mathbf{B}^{(3)}_{\text{SSPRK3}} = \begin{array}{c|ccc}
0 & 0 & 0 & 0 \\
1 & 1 & 0 & 0 \\
\frac{1}{2} & \frac{1}{4} & \frac{1}{4} & 0 \\
\hline
& \frac{1}{6} & \frac{1}{6} & \frac{2}{3}
\end{array}.
$$

For the fourth-order methods, we include the classic, 3/8-rule, and Ralston's Ralston (1962) methods, which are defined by:

$$
\mathbf{B}^{(4)}_{\text{classic}} = \begin{array}{c|cccc}
0 & 0 & 0 & 0 & 0 \\
\frac{1}{2} & \frac{1}{2} & 0 & 0 & 0 \\
\frac{1}{2} & 0 & \frac{1}{2} & 0 & 0 \\
1 & 0 & 0 & 1 & 0 \\
\hline
& \frac{1}{6} & \frac{1}{3} & \frac{1}{3} & \frac{1}{6}
\end{array}, \quad
\mathbf{B}^{(4)}_{\text{3/8-rule}} = \begin{array}{c|cccc}
0 & 0 & 0 & 0 & 0 \\
\frac{1}{3} & \frac{1}{3} & 0 & 0 & 0 \\
\frac{2}{3} & -\frac{1}{3} & 1 & 0 & 0 \\
1 & 1 & -1 & 1 & 0 \\
\hline
& \frac{1}{8} & \frac{3}{8} & \frac{3}{8} & \frac{1}{8}
\end{array},
$$

$$
\mathbf{B}^{(4)}_{\text{Ralston}} = \begin{array}{c|cccc}
0 & 0 & 0 & 0 & 0 \\
0.4 & 0.4 & 0 & 0 & 0 \\
0.45573725 & 0.29697761 & 0.15875964 & 0 & 0 \\
1 & 0.21810040 & -3.05096516 & 3.83286476 & 0 \\
\hline
& 0.17476028 & -0.55148066 & 1.20553560 & 0.17118478
\end{array}.
$$

The comparison results of various Butcher tableaux are shown in Tab. 5. The Heun's, Kutta's, and 3/8-rule variants achieve the best performance from the second-order to the fourth-order solver. Among the second-order to the fourth-order solvers, the fourth-order solver achieves the best performance, while the third-order solver slightly outperforms the second-order method.

## C.2 ABLATION STUDY ON SAMPLING STEPS AND NFES

Although the high-order approximation improves reconstruction performance, it introduces additional NFEs. For instance, our fourth-order RK solver with 30 sampling steps requires 240 NFEs, which is equivalent to a second-order method with twice the number of sampling steps. To rule out the possibility that the improvement is merely due to the increased NFEs, we conduct an ablation study on sampling steps and NFEs. The results are illustrated in Tab. 6. All experimental results indicate that increasing the number of sampling steps generally improves reconstruction performance.

Table 7: Ablation study on the manipulation of different decoupled attention components.

| Attention Map | | | | Attention Feature | | Structure | Unedited Fidelity | | CLIP Similariy | |
|---|---|---|---|---|---|---|---|---|---|---|
| $M_{CC}$ | $M_{II}$ | $M_{CI}$ | $M_{IC}$ | $V_C$ | $V_I$ | Distance↓ | PSNR ↑ | SSIM ↑ | Whole ↑ | Edited ↑ |
| ✗ | ✗ | ✗ | ✗ | ✗ | ✗ | 0.0606 | 19.01 | 0.7540 | 25.97 | 23.27 |
| ✓ | ✗ | ✗ | ✗ | ✗ | ✗ | 0.0610 | 19.02 | 0.7544 | 26.07 | 23.29 |
| ✗ | ✓ | ✗ | ✗ | ✗ | ✗ | 0.0333 | 22.68 | 0.8234 | 25.19 | 22.50 |
| ✓ | ✓ | ✗ | ✗ | ✗ | ✗ | 0.0332 | 22.70 | 0.8236 | 25.17 | 22.51 |
| ✗ | ✗ | ✓ | ✗ | ✗ | ✗ | 0.0606 | 19.04 | 0.7547 | 26.16 | 23.35 |
| ✗ | ✗ | ✗ | ✓ | ✗ | ✗ | 0.0573 | 19.42 | 0.7630 | 25.93 | 23.16 |
| ✗ | ✗ | ✓ | ✓ | ✗ | ✗ | 0.0573 | 19.40 | 0.7625 | 25.98 | 23.16 |
| ✓ | ✓ | ✓ | ✓ | ✗ | ✗ | 0.0372 | 22.78 | 0.8249 | 25.20 | 22.50 |
| ✗ | ✗ | ✗ | ✗ | ✓ | ✗ | 0.0589 | 19.28 | 0.7600 | 26.05 | 23.21 |
| ✗ | ✗ | ✗ | ✗ | ✗ | ✓ | 0.0246 | 25.35 | 0.8530 | 24.73 | 22.17 |
| ✗ | ✗ | ✗ | ✗ | ✓ | ✓ | 0.0245 | 25.36 | 0.8532 | 24.71 | 22.13 |

Table 8: Ablation study on manipulation methods. Here, $\mathcal{R}$ denotes the replacement operation, while $\mathcal{M}$ indicates the mean operation.

| Attention Map | | | | Attention Feature | | Structure | Unedited Fidelity | | CLIP Similariy | |
|---|---|---|---|---|---|---|---|---|---|---|
| $M_{CC}$ | $M_{II}$ | $M_{CI}$ | $M_{IC}$ | $V_C$ | $V_I$ | Distance↓ | PSNR ↑ | SSIM ↑ | Whole ↑ | Edited ↑ |
| $\mathcal{R}$ | ✗ | ✗ | ✗ | ✗ | ✗ | 0.0610 | 19.02 | 0.7544 | 26.07 | 23.29 |
| $\mathcal{M}$ | ✗ | ✗ | ✗ | ✗ | ✗ | 0.0609 | 19.02 | 0.7542 | 26.07 | 23.32 |
| ✗ | $\mathcal{R}$ | ✗ | ✗ | ✗ | ✗ | 0.0333 | 22.68 | 0.8234 | 25.19 | 22.50 |
| ✗ | $\mathcal{M}$ | ✗ | ✗ | ✗ | ✗ | 0.0412 | 21.36 | 0.8010 | 25.61 | 22.51 |
| ✗ | ✗ | $\mathcal{R}$ | ✗ | ✗ | ✗ | 0.0606 | 19.04 | 0.7547 | 26.16 | 23.35 |
| ✗ | ✗ | $\mathcal{M}$ | ✗ | ✗ | ✗ | 0.0609 | 19.02 | 0.7544 | 26.11 | 23.28 |
| ✗ | ✗ | ✗ | $\mathcal{R}$ | ✗ | ✗ | 0.0573 | 19.42 | 0.7630 | 25.93 | 23.16 |
| ✗ | ✗ | ✗ | $\mathcal{M}$ | ✗ | ✗ | 0.0591 | 19.22 | 0.7593 | 26.04 | 23.28 |
| ✗ | ✗ | ✗ | ✗ | $\mathcal{R}$ | ✗ | 0.0589 | 19.28 | 0.7600 | 26.05 | 23.21 |
| ✗ | ✗ | ✗ | ✗ | $\mathcal{M}$ | ✗ | 0.0598 | 19.13 | 0.7567 | 26.07 | 23.25 |
| ✗ | ✗ | ✗ | ✗ | ✗ | $\mathcal{R}$ | 0.0246 | 25.35 | 0.8530 | 24.73 | 22.17 |
| ✗ | ✗ | ✗ | ✗ | ✗ | $\mathcal{M}$ | 0.0311 | 23.38 | 0.8312 | 25.26 | 22.53 |

However, results on the vanilla RF and FireFlow Deng et al. (2025) reveal that such performance gains do not scale indefinitely. Notably, the reconstruction performance degrades significantly when the number of sampling steps reaches 120. In addition, although the NFEs of vanilla RF with 120 sampling steps, RF-Solver Wang et al. (2025) with 60 steps, FireFlow with 120 steps, our third-order RK solver with 40 steps, and our fourth-order RK solver with 30 steps are all about 240, our fourth-order method significantly outperforms the others. This demonstrates that the improvement in reconstruction performance is not solely owing to the increased NFEs.

## C.3 Ablation Study on Decoupled Attention Components

We conduct a comprehensive experiment to evaluate the influence of each attention region on the trade-off between fidelity and editability. In this experiment, we employ the third-order RK solver (Kutta's variant) with 8 sampling steps as the sampler, and apply attention replacement only to the single-stream DiT blocks at the first timestep. From the results shown Tab. 7, we draw the following conclusions: (1) For self-attention maps, replacing $M_{CC}$ yields only a very small improvement in both fidelity and editability, whereas replacing $M_{II}$ significantly enhances fidelity at the cost of reduced editability. (2) Manipulating the two types of cross-attention maps leads to different effects, *i.e.*, replacing $M_{CI}$ improves the editability while slightly enhancing the fidelity, whereas replacing $M_{IC}$ improves the fidelity but slightly reduces the editability. (3) For the value attention features, replacing $V_C$ slightly improves the fidelity while maintaining the editability, whereas replacing $V_I$ significantly enhances the fidelity but substantially reduces the editability. Therefore, the order of contributions to fidelity is: $V_I > M_{II} > M_{IC} > M_{CI} > M_{CC}$, while the order of contributions to editability is: $M_{CI} > M_{CC} \approx V_C > M_{II} > V_I$.

Table 9: Computational cost of the proposed RK Solver.

| Method | Order | NFEs | Runtime | GPU Memory |
|---|---|---|---|---|
| Vanilla RF | 1 | 60 | 45.9 | 35.51 |
| RK Solver | 2 | 120 | 91.5 | 35.51 |
| RK Solver | 3 | 180 | 136.9 | 35.51 |
| RK Solver | 4 | 240 | 182.6 | 35.51 |

Table 10: Computational cost of the proposed DDTA. Here, $\mathcal{R}$ denotes the replacement operation, while $\mathcal{M}$ indicates the mean operation.

| Attention Map | | | | Attention Feature | | Runtime | GPU Memory |
|---|---|---|---|---|---|---|---|
| $M_{CC}$ | $M_{II}$ | $M_{CI}$ | $M_{IC}$ | $V_C$ | $V_I$ | | |
| ✗ | ✗ | ✗ | ✗ | ✗ | ✗ | 35.6 | 35.51 |
| $\mathcal{R}$ | ✗ | ✗ | ✗ | ✗ | ✗ | 55.4 | 36.89 |
| $\mathcal{M}$ | ✗ | ✗ | ✗ | ✗ | ✗ | 55.3 | 36.89 |
| ✗ | $\mathcal{R}$ | ✗ | ✗ | ✗ | ✗ | 108.7 | 38.39 |
| ✗ | $\mathcal{M}$ | ✗ | ✗ | ✗ | ✗ | 111.8 | 38.39 |
| ✗ | ✗ | $\mathcal{R}$ | ✗ | ✗ | ✗ | 56.3 | 36.89 |
| ✗ | ✗ | $\mathcal{M}$ | ✗ | ✗ | ✗ | 56.3 | 36.89 |
| ✗ | ✗ | ✗ | $\mathcal{R}$ | ✗ | ✗ | 56.3 | 36.89 |
| ✗ | ✗ | ✗ | $\mathcal{M}$ | ✗ | ✗ | 56.3 | 36.89 |
| ✗ | ✗ | ✗ | ✗ | $\mathcal{R}$ | ✗ | 51.6 | 36.89 |
| ✗ | ✗ | ✗ | ✗ | $\mathcal{M}$ | ✗ | 51.8 | 36.89 |
| ✗ | ✗ | ✗ | ✗ | ✗ | $\mathcal{R}$ | 51.7 | 36.89 |
| ✗ | ✗ | ✗ | ✗ | ✗ | $\mathcal{M}$ | 51.9 | 36.89 |

### C.4 Ablation Study on Manipulation Method

We conduct a series of experiments to evaluate the effectiveness of different manipulation strategies. Experimental results in Tab. 8 show that the mean operation is slightly less effective than the replacement operation. Therefore, users can precisely control the edited image by customizing the manipulation method, the number of DiT blocks, and the sampling steps to achieve the most satisfactory results.

## D Appendix: Discussion on Computational Cost

Although our proposed framework achieves substantial improvements in both reconstruction and editing performance, it inevitably incurs additional computational overhead. Here, we present a quantitative analysis of the computational cost incurred by the proposed RK Solver and DDTA, reporting the runtime (in seconds per image) and GPU memory usage. In this evaluation, all experiments are conducted on a single NVIDIA L40 GPU, and the data is running under the bfloat16 floating-point format. The results of RK Solver, as shown in Tab. 9, lead to the following observations: (1) since the proposed method is training-free, all solvers occupy the same GPU memory, and (2) the runtime increases approximately linearly with the solver's order, as higher-order solvers require more NFEs. To quantify the computational overhead introduced by DDTA, we evaluate the runtime and GPU memory consumption under different attention manipulation strategies. As shown in Tab. 10, applying DDTA leads to additional computational cost in both runtime and GPU memory usage. This overhead is directly correlated with the dimensionality of the preserved attention maps or features, following the order: $M_{II} > M_{CI} = M_{IC} > M_{CC} > V_I > V_C$. It is worth noting that the additional cost originates from storing and reusing attention maps/features from the inversion branch. Therefore, the specific manipulation type (e.g., replace or mean) does not affect the computational overhead.

# E    APPENDIX: LLMs USAGE STATEMENT

In this paper, we utilized LLMs solely for the purpose of language polishing and stylistic refinement of the manuscript. Specifically, the LLM was employed to optimize the clarity, fluency, and consistency of the written English expression to enhance the readability. Notably, the LLMs did not play any role in research ideation, experimental design, or formulation of key conclusions of this study.

