# OpenReview forum: "Runge-Kutta Approximation and Decoupled Attention for Rectified Flow Inversion and Semantic Editing"
_ICLR.cc/2026/Conference — ICLR 2026 Conference Withdrawn Submission_

### Official Review · Reviewer_VirA · 2025-10-25

**Soundness:** 3
**Presentation:** 3
**Contribution:** 2
**Rating:** 4
**Confidence:** 5

**Summary:**

This paper aims to utilize the Rectified-flow model for image editing, which proposes two contributions to improve rectified flow (RF) models: (1) introducing a high-order Runge-Kutta (RK) solver for more accurate inversion, and (2) proposing Decoupled Diffusion Transformer Attention (DDTA) to disentangle text and image modalities for more controllable semantic editing in multimodal diffusion transformers. Extensive experiments such as image reconstruction and text-guided semantic editing are demonstrated to illustrate the effectiveness of the proposed method.

**Strengths:**

- Application of Runge-Kutta solvers enhances the image inversion fidelity in rectified flow models. This is substantiated both theoretically (Appendix A) and empirically (Tables 1, 5, 6, and 9)
- The DDTA separates text/image attention within MM-DiT architectures, facilitating fine-grained semantic editing.
- Experiments on image editing and reconstruction tasks illustrate the effectiveness of the proposed methods.
- The paper is clear and well-written.

**Weaknesses:**

- The novelty of this paper is limited. Using the RK method for solving the rectified flow ODE has been explored in the original Rectified Flow paper [1]. Utilizing feature injection to enhance the consistency between source and edited images is also extensively explored in previous works [3, 4]. There are also several missed baselines: FlowEdit [2] and QK-Edit [3].
- The qualitative results are not enough. As we know, in the field of image editing, qualitative results are an important way to demonstrate the effectiveness of the method. The authors should provide more qualitative results in both main results and ablation study.
- The impact of feature injection step in DDTA is not clear. As illustrated in RF-Edit [5] Figure 7, the feature injection step could significantly impact the results of image editing. Under different feature injection steps, could DDTA performs consistently better compared with baseline methods such as RF-Edit and QK-Edit? Authors are encouraged to provide more results on this.



[1] Flow Straight and Fast: Learning to Generate and Transfer Data with Rectified Flow, ICLR 2023.

[2] FlowEdit: Inversion-Free Text-Based Editing Using Pre-Trained Flow Models, ICCV 2025.

[3] QK-Edit: Revisiting Attention-based Injection in MM-DiT for Image and Video Editing, ICCV 2025.

[4] DiTCtrl: Exploring Attention Control in Multi-Modal Diffusion Transformer for Tuning-Free Multi-Prompt Longer Video Generation, CVPR 2025.

[5] Taming Rectified Flow for Inversion and Editing, ICML 2025.

**Questions:**

See weakness.

---

### Official Review · Reviewer_ghif · 2025-10-27

**Soundness:** 2
**Presentation:** 2
**Contribution:** 2
**Rating:** 4
**Confidence:** 5

**Summary:**

This paper presents a two-pronged approach to address critical limitations in applying Rectified Flow (RF) models, particularly those using Multimodal Diffusion Transformer (MM-DiT) backbones, to real-world image editing.

First, to improve the low inversion accuracy that compromises image fidelity, the authors propose the Runge-Kutta (RK) Solver. This is an efficient, high-order inversion method based on the established Runge-Kutta solver for differential equations, designed to provide a more accurate approximation of the RF's ODE trajectory.

Second, to tackle the challenge of entangled multimodal attention in MM-DiTs, which hinders precise semantic control, the paper introduces Decoupled Diffusion Transformer Attention (DDTA). DDTA is a mechanism that disentangles the text and image attention features, enabling fine-grained manipulation during the editing process.

The combination of the RK Solver and DDTA is shown through extensive experiments on reconstruction and semantic editing tasks to achieve state-of-the-art performance in terms of both fidelity (e.g., up to 2.39 dB PSNR gain in reconstruction) and editability compared to existing RF and DDIM-based methods.

**Strengths:**

High-Fidelity Inversion via RK Solver: The proposed RK Solver successfully incorporates established high-order numerical methods into the RF framework, achieving significantly improved reconstruction fidelity (up to 25.68 PSNR for $r=4$ variant, a substantial gain over existing methods like FireFlow). This addresses a core bottleneck for inversion-based editing in RF models.

Principled Decoupling for MM-DiT: The DDTA mechanism provides a necessary and well-motivated solution for attention manipulation in modern MM-DiT architectures (e.g., FLUX). By explicitly decoupling entangled text-image attention maps, it enables fine-grained control that was previously only viable in UNet-based models with separate attention components (like Prompt-to-Prompt).

Efficiency in Editing: The method demonstrates superior overall editing performance (fidelity and editability) with significantly fewer sampling steps (e.g., 5-8 steps) compared to most baselines (e.g., 25-50 steps). This highlights the computational efficiency gained by the higher-order RK solver combined with effective attention manipulation.

Comprehensive Evaluation: The paper includes thorough quantitative and qualitative results, including ablation studies on the RK solver order and Butcher tableau selection (Table 1, Table 5) and the detailed contribution of each decoupled attention component (Table 7), providing deep insights into the method's design choices. The MLLM-based user study (Table 3) further strengthens the claim of superior editing quality.

**Weaknesses:**

Significant Computational and Memory Overhead (Critical): While high-order solvers improve accuracy, they inherently increase the Number of Function Evaluations (NFEs) and thus runtime linearly (Table 9). Furthermore, DDTA, by storing and manipulating attention maps, introduces notable GPU memory consumption, which is a practical limitation for large-scale models and limited hardware (Table 10). The paper adequately acknowledges this but the extent of this overhead (e.g., 4x runtime for r=4 RK Solver, and memory increase for DDTA) remains a major concern for practical deployment.

Lack of Robustness Analysis for Butcher Tableaux: Although an ablation on different Butcher tableaux is provided (Table 5), the results show that only a few specific variants (Heun's, Kutta's, 3/8-rule) perform optimally, while others degrade significantly. This sensitivity indicates that the RK Solver's performance is not universally robust across RK variants, and requires non-trivial empirical tuning to find the optimal configuration for a given model, which undermines its claimed generality.

Limited Exploration of Editing Operations in DDTA: The DDTA manipulation primarily focuses on simple Replacement or Mean operations on the attention maps/features. Given the fine-grained control enabled by the decoupling, the potential for more advanced, prompt-driven manipulation strategies (e.g., cross-attention injection based on prompt differences) is largely unexplored. The current implementation, while effective, feels foundational rather than fully exploiting the DDTA's capabilities.

Missing Comparison on Efficiency/Cost Trade-off: The paper lacks a focused experiment directly comparing the fidelity/editability gain per unit of computational cost (e.g., normalizing performance metrics by NFEs or runtime against baselines). While it mentions superior performance with fewer steps, a step-for-step comparison normalized by the increased NFE/runtime would provide a fairer assessment of its true efficiency advantage.

**Questions:**

Above

---

### Official Review · Reviewer_a6NH · 2025-10-31

**Soundness:** 3
**Presentation:** 3
**Contribution:** 2
**Rating:** 4
**Confidence:** 3

**Summary:**

The paper proposes to address two primary challenges in using RF models for real-world image manipulation: low inversion accuracy and entangled multimodal attention. The framework contains two main components: (1 )Inversion with Runge-Kutta Solver which is an efficient higher-order inversion method based on the RK differential equation solver. This improves inversion fidelity and image reconstruction. (2) Decoupled Diffusion Transformer Attention (DDTA) which disentangles text and image attention in MM-DiT, and enables more precise semantic control in test guided image editing. Experiment results demonstrate SoTA results of the proposed methods.

**Strengths:**

1. The authors provides thorough experiment analysis and ablation studies.
2. The method is shown to achieve the best overall performance with significantly fewer sampling steps (e.g., 5 to 8 steps in editing tasks) compared to many baselines, though at cost of computational overhead.
3. Both qualitative and quantitative results are good.

**Weaknesses:**

Though the proposed method has theoretical contribution, practically, the method largely increases computational overhead. The high-order RK Solver introduces additional computational cost, and the DDTA component incurs notable memory consumption (Tab. 9, 10). Besides, the design choices that boost fidelity (e.g., $V_I$ or $M_{II}$ in tab. 7) reduce editability, suggesting the best settings require careful balancing for different tasks.

**Questions:**

1. The attention manipulation only happens at the first sampling step. How did the authors determine the first step is optimal, and what are the specific benefits or drawbacks of extending this manipulation to subsequent steps or different blocks? (L939)
2. Since the fourth-order RK solver has the highest computational cost (240 NFEs at 30 steps) but the best reconstruction performance (Tab. 6, Tab. 9), what is the practical recommended trade-off (order and steps) for real-world applications where runtime and efficiency are primary concerns?

---

### Note · Authors · 2025-12-04

I have read and agree with the venue's withdrawal policy on behalf of myself and my co-authors.